# Southern Europe and Western Asia Marine Heat Waves (SEWA-MHWs): a dataset based on macroevents

Giulia Bonino[1*], Simona Masina[1], Giuliano Galimberti[2], and Matteo Moretti[1,2]

[1]Ocean Modeling and Data Assimilation Division, Fondazione Centro Euro-Mediterraneo sui Cambiamenti Climatici, Bologna, Italy.
[2]Department of Statistical Sciences, University of Bologna, Bologna, Italy

**Correspondence:** Giulia Bonino (giulia.bonino@cmcc.it)

**Abstract.** Marine Heat Waves (MHWs) induce significant impacts on marine ecosystems. There is a growing need for knowledge about extreme climate events to better inform decision-makers on future climate-related risks. Here we present a unique observational dataset of MHWs macroevents and their characteristics over the Southern Europe and Western Asian (SEWA) basins, named SEWA-MHWs dataset (Bonino et al., 2022). The SEWA-MHWs dataset is derived from the European Space Agency (ESA) Climate Change Initiative (CCI) Sea Surface Temperature (SST) v2 dataset and it covers the 1981-2016 period. The methodological framework used to build the SEWA-MHWs dataset is the novelty of this work. Firstly, the MHWs detected in each grid point of the ESA CCI SST dataset are relative to a time-varying baseline climatology. Since intrinsic fluctuation and anthropogenic warming are redefining the mean climate, the baseline considers both the trend and the time-varying seasonal cycle. Secondly, using a connected component analysis, MHWs connected in space and time are aggregated in order to obtain macroevents. Basically, a macroevent-based dataset is obtained from a grid cell-based dataset without losing high resolution (i. e. grid cell) information. SEWA-MHWs dataset can be used for many scientific applications. For example, we identified phases of the well known MHW of summer 2003 and, taking advantage of statistical clustering methods, we clustered the largest macroevents in SEWA basins based on shared metrics and characteristics.

## 1 Introduction

Over the past decades, anomalous prolonged events of warm sea surface water, known as marine heatwaves (MHWs), have developed globally, both in the open ocean and in coastal regions, leading to serious consequences for marine ecosystems. The ecosystems are very sensitive to abrupt temperature changes and they can reach their "tipping point" (Lenton et al., 2008; Serrao-Neumann et al., 2016), which means entering into an unknown state which may be completely distant from the previous one. Increased ocean temperatures imply a large number of ecological repercussions spanning from a coastwide onset of toxic algae (McCabe et al., 2016; Ryan et al., 2017) to dramatic range shifts of species at all trophic levels (Cavole et al., 2016; Sanford et al., 2019). These adverse conditions during MHWs can lead to substantial economic losses for important fisheries and aquaculture industries (McCabe et al., 2016; Cavole et al., 2016; Frölicher, 2019).

Some relevant examples of unprecedented ocean temperature anomalies are the 2011 Western Australia marine heatwave in the eastern Indian Ocean (Pearce et al., 2011), and the persistent 2014–2016 "Blob" in the North Pacific (Di Lorenzo and

Mantua, 2016). In the Mediterranean Sea, several works reported an anomalous warm sea surface temperature during the summer of 2003 (e.g., Olita et al., 2007; Marullo and Guarracino, 2003; Grazzini and Viterbo, 2003). Darmaraki et al. (2019) identified the MHWs of 2003, 2012, and 2015 as the basin-scale most severe surface events during the 1982–2017 period. Marbà et al. (2015), focusing on impacts of these extreme events on Mediterannean biota, reported basin-scale MHWs during 1994 and 2009 and a regional MHW over the Adriatic, the Ionian and parts of the Levantine basin during the 1998. Other studies on ecological impacts identified MHWs over the western Mediterranean Sea during 2008 (Cebrian et al., 2011) and 2006 (Kersting et al., 2013), and over Adriatic Sea during 2009 (Di Camillo et al., 2013). Very limited information is available for MHWs in the Black Sea and in the Caspian Sea (e.g. Mohamed et al. (2022)). They are usually related to works where MHWs are detected and studied at global scales, such as Holbrook et al. (2020) and Sen Gupta et al. (2020). A basic definition of MHWs (i.e., IPCC SROCC) states they are extremely anomalous temperatures in the ocean (Pörtner et al., 2019). However, to compare events and study their impacts, they must also be identifiable, with clear start and end dates and measurable characteristics. Hobday et al. (2016) were the first to propose a definition for MHWs, according to which the temperature must be higher than a given percentile (e.g., 90th, relevant to a reference climatology) and must persist for at least five days. This definition has been widely adopted by the oceanographic community (e.g., Holbrook et al., 2019, Oliver et al., 2021, Smale et al., 2019). However, it is worth mentioning that this definition is characterised by flexibility in the choice of the set-up parameters (such as the climatology and the percentile threshold), thus limiting comparability among studies.

Most of the research conducted in this emerging field exploits this definition to study extreme events at individual locations (i.e. grid cells). Nevertheless, the grid cell-based MHWs events are likely connected in time and in space being part of the same extreme macroevent. Very few studies put effort into defining macroevents. For example, Darmaraki et al. (2019) and Pastor and Khodayar (2022) define the spatiotemporal extent of the MHW when a minimum of 20% and 5% of the Mediterranean basin is affected by grid cell-based MHWs, respectively. Sen Gupta et al. (2020) used a semi-objective procedure to characterize and to detect globally the most extreme MHWs macroevents and Woolway et al. (2021) used a connected component analysis to study extreme temperature macroevents in the Laurentian Great Lakes. More recently, Sun et al. (2022) tracked the evolution in time of constructed snapshots of spatially compact MHWs. Macroevent-based studies facilitate the investigation of MHWs drivers (Sun et al., 2022), which are currently not well understood (Holbrook et al., 2020). Driving mechanisms, which are usually seasonal and location-dependent, are related to oceanic and atmospheric forcing or a combination of both (e.g., Oliver et al., 2018, 2021; Holbrook et al., 2019; Frölicher and Laufkötter, 2018). Examples of key phenomena which cause these extreme temperatures are anomalous horizontal advection, anomalous heat fluxes, sea level pressure anomalies, reduced coastal upwelling, Ekman pumping or the re-emergence of warm anomalies from the subsurface (Schlegel et al., 2021; Holbrook et al., 2019, 2020). The time scale of these relevant physical drivers and processes involved in MHW emergence spans from days (e.g. anomalous heat fluxes), to weeks (e.g. blocking systems and atmospheric teleconnections), to months (e.g. re-emergence of warm anomalies from the subsurface) and years (e.g. climate modes and oceanic teleconnections) (e.g., Oliver et al., 2018, 2021).

Given the pronounced warming trend in recent years, Benthuysen et al. (2020) and Holbrook et al. (2020) suggest the need for a more comprehensive and consistent framework to report MHWs. To the best of our knowledge, there is no available

MHWs macroevents dataset in the literature. Even thought MHWs are derived from Sea Surface Temperature (SST), whose observations are available, the processing to detect MHWs is computational and/or time demanding, especially for high resolution SST data.

In short, the current state of knowledge about MHWs requires addressing the need for more comprehensive efforts to document and to report these extreme events. The aim of this paper is to provide a unique dataset of MHWs macroevents derived from the European Space Agency (ESA) Climate Change Initiative (CCI) Sea Surface Temperature v2 dataset. The dataset consists in a daily dataset of MHWs macroevents and their characteristics over the Southern Europe and West Asian (SEWA) basins, named as SEWA-MHWs dataset. We have focused on SEWA basins because they represent a well known "Hot Spot" region for climate change (Giorgi, 2006) and, in particular, for this specific phenomenon (Garrabou et al., 2009; Giorgi, 2006; Cramer et al., 2018; Pastor and Khodayar, 2022; Pastor et al., 2020; Garrabou et al., 2022; Ciappa, 2022). Marine heatwaves caused unprecedented biological impacts, especially in the Mediterranean Sea (Garrabou et al., 2022; Cramer et al., 2018; Marbà et al., 2015; Rivetti et al., 2014), seriously affecting marine biodiversity (Juza et al., 2022). Moreover, the Mediterranean Sea is recognized as an exemplary model for assessing the ecological and biological impacts of climate change (Garrabou et al., 2022; Cramer et al., 2018).

We generated the SEWA-MHWs dataset in a new consistent framework. In brief, we detected MHWs relative to a time-varying baseline climatology in each grid point of the ESA CCI SST dataset, then, using a connected component analysis, we aggregated the spatiotemporally connected MHWs in order to obtain macroevents.

This paper is organized as follows: in Section 2, we present the data used to produce the SEWA-MHWs dataset and the studied area. In section 3 we describe the methodological framework applied to obtain the dataset and in section 4 we offer an example of its scientific application. Section 5 reports the availability of codes and data used to build SEWA-MHWs dataset. Our conclusions and outlook of the work are summarized in Section 6.

## 2 ESA SST CCI data and Study Area

To generate the SEWA-MHWs dataset we used the European Space Agency (ESA) Climate Change Initiative (CCI) SST dataset v2.1 (hereinafter ESA CCI SST dataset). This dataset, available in the CEDA catalogue [1], provides global daily satellite-based SST data covering the period from September 1981 to December 2016. A detailed overview of processing updates, and of the history behind, for the ESA CCI SST dataset v2.1 is presented by Merchant et al. (2019). The ESA CCI SST v2.1 dataset is designed to provide a long-term, stable, low-bias climate data record derived from different infrared sensors, i.e., the AVHRR, (A)ATSR and SLSTR series of sensors (Merchant et al., 2019, 2014). 17 missions from 1981 to 2016 contributed to the ESA CCI SST dataset (e.g. NOAA-6, ATSR-1, Metop-A) and they are fully described in Merchant et al. (2019) (see their Figure 3). Different processing levels of ESA CCI SST dataset are available: single-sensor data on the native swath grid (Level-2); uncollated single-sensor (Level-3U); collated multi-sensor (Level-3C) gridded data; and optimally interpolated (Level-4) multi-sensor data. Here, only the spatially complete Level-4 product (ESA CCI SST L4), obtained through the Operational

---

[1]https://catalogue.ceda.ac.uk/uuid/62c0f97b1eac4e0197a674870afe1ee6

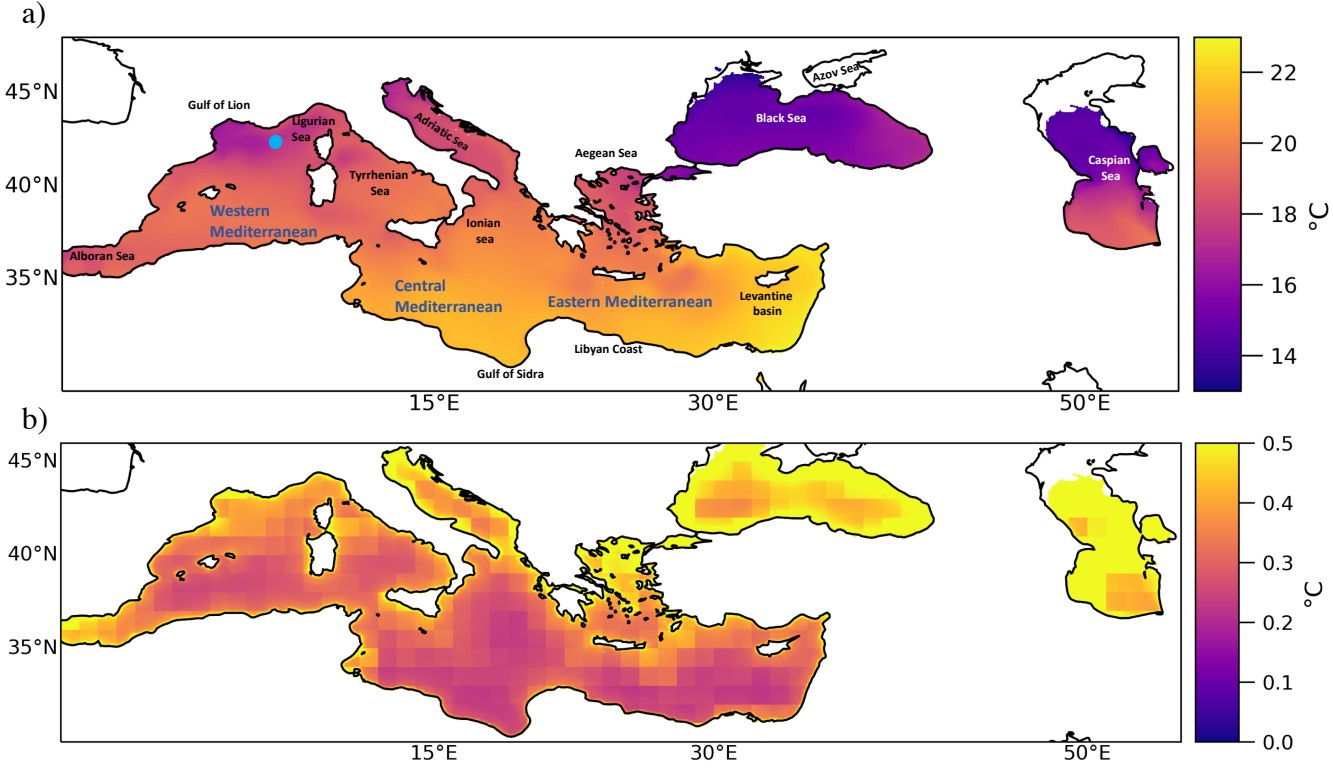

**Figure 1.** a) Mean SST climatology detected by STL-method with geographical names. Blue circle identifies the Western Mediterranean location of the time-series shown in Figure 4; b) Mean SST uncertainties during the studied period.

Sea Surface Temperature and Sea Ice Analysis (OSTIA) system (Donlon et al., 2012), is considered. This dataset consists of daily maps of average SST at 20 cm nominal depth with 0.05° x 0.05° of horizontal resolution, covering the period from September 1981 to December 2016. The ESA CCI SST L4 is adjusted to 20cm depth to be comparable with drifter and historic bucket temperature measurements. Moreover, the dataset is also adjusted in time to address the SST diurnal heating

allowed by different overpass times of satellites that differentially sample SST. In particular, the temporal adjustment is applied as an estimate of the change in SST between the observation time and the nearest of 10.30 or 22.30 local mean solar time, which is a good approximation of the SST daily mean (Morak-Bozzo et al., 2016). Observation data from 1991 onwards needed only minimal adjustment, being always available a mid-morning satellite observation. Therefore, the diurnal heating is

somehow taken into account in the ESA SST CCI L4 data processing. The data are adjusted in depth and in time to be more representative of the daily SST mean, the meaningful data frequency to define MHW events. The neglected diurnal warming in the SST dataset (e.g SST provided at the foundation depth or SST provided each day at the nominal time 00:00 UTC) could have otherwise compromised the estimation of the extreme events (Marullo et al., 2016). It is worth mentioning that passing from Level-2 to Level-4 degrades resolution and increases uncertainties, slightly compromising the detection of extremes,

especially the most geographically localized ones. Nevertheless, the interpolated, gap-filled L4 analysis is perfectly suitable for our purpose to archive and describe MHWs over the SEWA region. In particular, ∼5 km of horizontal resolution of ESA CCI SST L4 is in line with other satellite-based datasets (e.g. OSTIA, Stark et al., 2007) and with state-of-art regional ocean models outputs (Clementi et al., 2021) and, in addition, the continuity in time and in space guaranteed by this dataset is needed to define spatiotemporally connected MHWs. In this work we focused on the Southern Europe and West Asian basins, namely we constructed our SEWA-MHWs dataset over the Mediterranean Sea, the Black Sea, and the Caspian Sea (see Figure 1a). It is worth mentioning that the ESA CCI SST L4 mean uncertainties over all the studied period ranges, in the Mediterranean Sea, from 0.1 to 0.25° in the open ocean and around 0.5° along the coast. Over the Black Sea and the Caspian Sea, the uncertainties exceed 0.3° in the open ocean and 0.5° along the coast (Figure 1b). In particular, the largest uncertainties in the dataset occur during the 1980s, reflecting the deficiencies of in situ observations in space and in time at that time, and the fact that only one AVHRR sensor at a time was available (Merchant et al., 2019).

## 3   Methodological Framework

In this Section we present the methodological framework developed to obtained the SEWA-MHWs dataset (Bonino et al., 2022). The flow diagram in Figure 2 illustrates and summarizes the data processing to generate the SEWA-MHWs dataset, it also highlights the keywords used in the following paragraphs. We first detected the grid cell-based MHWs applying a new baseline climatology estimation strategy and we detected the MHWs metrics (Section 3.1). Then, we identified spatiotemporally connected MHWs to define MHWs macroevents (Section 3.2). The MHWs metrics, the MHWs macroevents together with some relevant atmospheric variables, taken from ERA5 dataset, form the SEWA-MHWs dataset (Section 3.3). In this Section we also evaluated our method describing a well know macroevent in comparison with the literature (Section 3.2.1).

### 3.1   Grid point MHWs definition and their characteristics

We identified MHWs for each grid point of ESA CCI SST dataset following the definition of Hobday et al. (2016): "MHWs are identifiable events with start and end dates, a persistent duration of at least five days and anomalous warmer sea surface water relative to a threshold (90th or 99th percentile) in a 30-year baseline climatology". To build a globally consistent detection framework to study MHWs drivers and variability, it is therefore crucial the choice of a proper baseline climatology, along with the choice of the percentile threshold. To describe and to study MHWs it is also fundamental to define MHWs metrics and characteristics. In short, from the analysis presented in the next paragraphs we detected MHWs for each ∼5 km grid for the ESA SST CCI dataset and we obtained daily maps of MHWs occurrence, i.e. binary maps identifying which of the ∼5 km grid points across the ocean experienced a MHW (taking the value 1 for grid points experiencing a MHW and 0 otherwise), and daily maps of MHWs caracteristics (Figure 2).

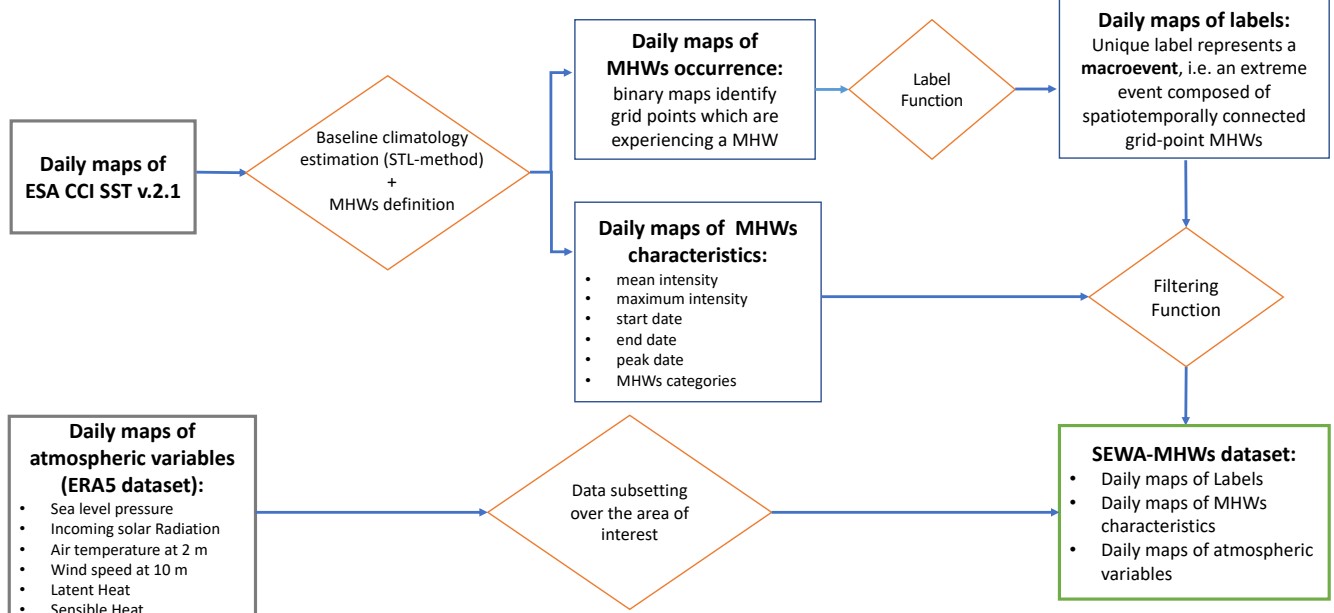

**Figure 2.** Flow diagram of the data processing to generate the SEWA-MHW dataset. Gray squares indicate data input; Blue squares indicate intermediate or output data; green square indicates the final output (SEWA-MHWs dataset); orange rhombuses indicate the function/process applied.

### 3.1.1 Baseline climatology estimation and threshold

Hobday et al. (2016) estimate the baseline climatology as a seasonally varying climatology calculated over 30-year period without removing long-term trend (hereinafter Hobday-method, see Hobday et al., 2016 for details). In contrast, we considered trend and time-varying seasonality in the baseline climatology estimation. As the definition states, MHWs detection depends on the underlying SST properties (Oliver et al., 2021). Recent studies show that increases in both the mean SST and the variability of SST due to global warming can lead to increase in warm temperature extremes (Pierce et al., 2012), so that by the late twenty-first century most of the global ocean will reach a permanent MHW state (Oliver et al., 2018; Holbrook et al., 2020; Frölicher et al., 2018). Persistent warming of SST and the increased frequency and intensity of extreme events indicate that the global ocean is experiencing unprecedented climate normals (Tanaka and Van Houtan, 2022). Thus, the rationale of considering trend and time-varying seasonality in the baseline climatology estimation is to take into account that the mean state of the ocean is changing over time due to natural variability and anthropogenic climate change.

We estimated the baseline climatology using a non-parametric time series decomposition algorithm, named Seasonal-Trend Decomposition Procedure Based on LOESS (hereinafter STL-method), designed and described by Cleveland et al. (1990). It is based on the non-parametric technique known as Locally Estimated Scatterplot Smoothing (LOESS), also commonly called Local Regression. The method estimates time-varying trend and seasonality for each time series. The STL algorithm that we

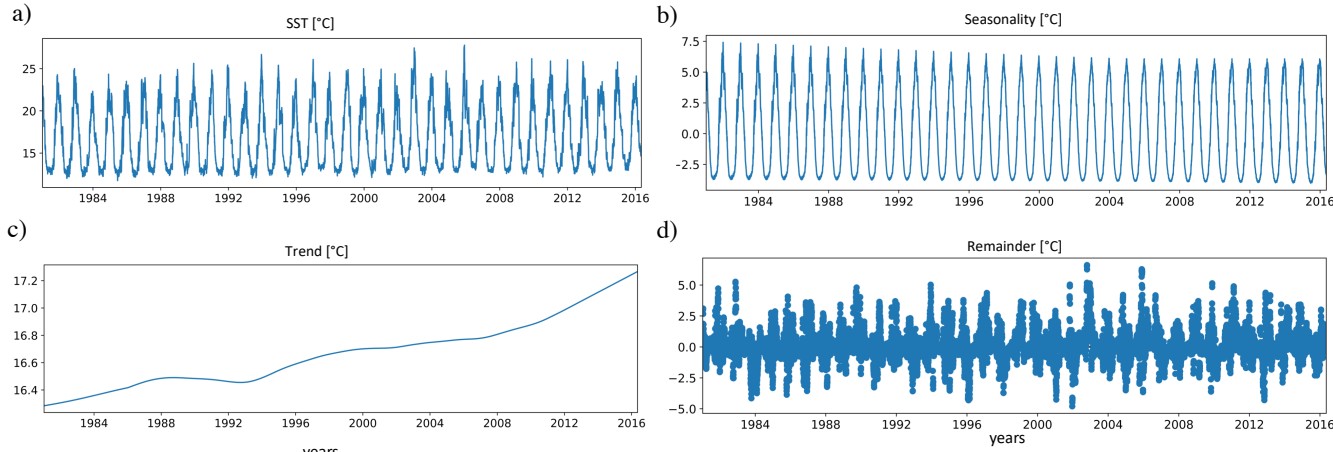

**Figure 3.** STL decomposition for a SST times-series in Western Mediterranean (blue circle in Figure 1a): a) SST times-series, b) Time-varying seasonality, c) Trend, d) Residuals.

applied to ESA SST CCI is implemented in a freely available Python function named STL [2] from the Statsmodel Python
package [3]. Figure 3 shows the STL-method decomposition for an ESA SST CCI time series in the western Mediterranean (blue
circle in Figure 1a). The main parameters to set for the STL algorithm are related to the periodicity of the seasonal signal to
be extracted, and to some LOESS smoothing parameters for the trend and for the seasonality. We tuned our estimation of the
trend with a smoothing window of 10 years. This choice produces a trend which captures both the long-term trend and low
frequency variability (Figure 3c). Regarding the seasonal cycle, we tuned its estimation with a yearly periodicity, smoothed
over 5 years (Figure 3b). The time-varying amplitude of the seasonality captures increasing/decreasing trends in the seasonal
variability of the time series. The sum of the trend and the seasonality is considered as our baseline climatology (Figure 4a,
orange line).

The STL-method climatology (orange line, Figure 4a), in contrast to the Hobday-method climatology (black line, Figure 4a),
varies with location, due to the trend, and in variability, due to the time-varying seasonality. The Hobday-method climatology
results instead in a pure periodic seasonal climatology with a flat trend. In particular, the time series of the differences between
the two climatologies in Figure 4b shows an increased seasonality of STL-method climatology. The STL-climatology is higher
(lower) with respect to Hobday climatology during summer (winter) season. The differences between climatologies are maxima
during 1983-1992 period for the winter season, while during the 2012-2016 period the summer discrepancies reach 2°C due to
the fact that STL-method includes an increased trend in the estimation of climatology. It is evident that SST time series could
present different time evolution. They could show decreasing, oscillating or stationary trends in the mean and in the variance.
One of the main advantages of the STL-method decomposition is that it allows to work on SST time series in a consistent

---

[2]https://www.statsmodels.org/devel/examples/notebooks/generated/stl_decomposition.html
[3]https://www.statsmodels.org/stable/index.html

framework, where the residuals, obtained by substracting the estimated trend and climatology from the corresponding observed SST values, are the relevant comparable series, and trend and seasonality form together the time-varying baseline climatology.

As suggested by Hobday et al. (2016), we computed the 90th percentile of the residuals and used it as threshold in order to
170 detect grid point MHW and to obtain daily maps of MHWs occurence (Figure 2). In addition, two successive MHW events with a 2-day or less time break were considered as a single continuous events.

It is worth noting that we did not consider grid cells where sea ice can be present, so that MHWs are not available over the North Caspian Sea and the Azov Sea (Figure 1).

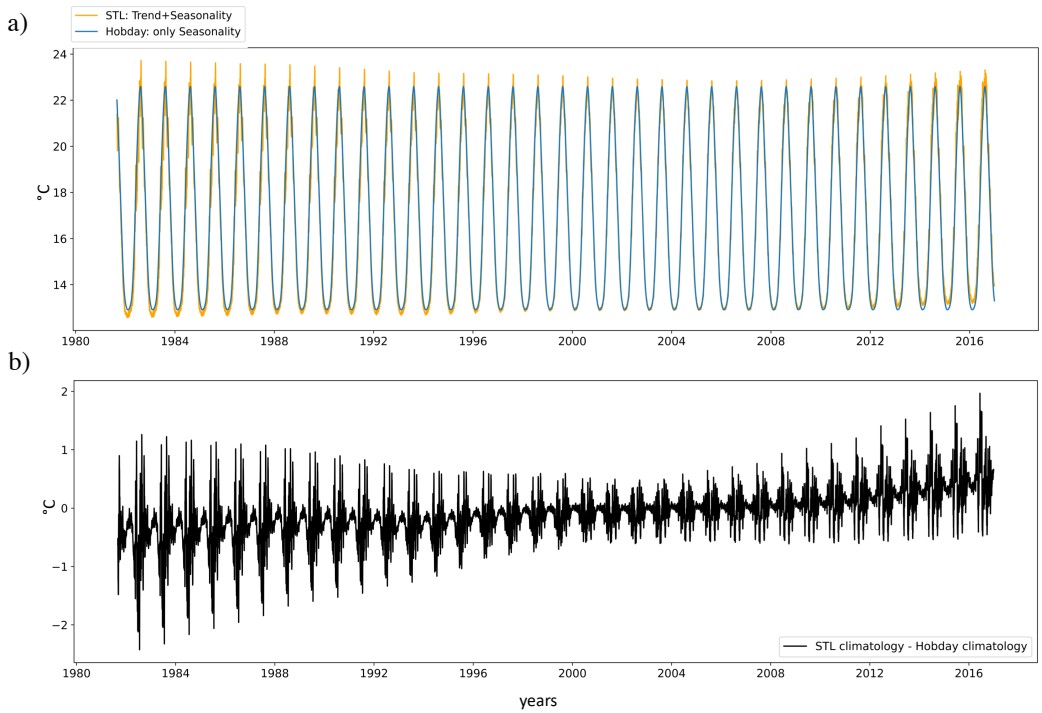

**Figure 4.** a) STL mean climatology (orange line) and Hobday mean climatology (blue line) for a SST times-series in Western Mediterranean (blue circle in Figure 1), b) Differences between STL mean climatology (orange line) and Hobday mean climatology (blue line) in a)

### 3.1.2 MHWs characteristics

Following the approach of Hobday et al. (2016), for each detected MHW we calculated the following MHWs metrics and characteristics, generating daily maps of MHWs characteristics (Figure 2): mean intensity and maximum intensity (i.e. the average and maximum temperature anomaly over the duration of the event), the start date, the end date and the peak date (i.e. indices at start, end and at the peak of the marine heatwave in times series of origin). In addition, each detected MHWs was assigned to a category describing its severity (Hobday et al., 2018). These categories range from 1 to 4 and they are based on

the maximum intensity in multiples of the 90th percentile exceedances, i.e., category 1 indicates the MHW peak intensity is >=1 times the value of the 90th percentile threshold, but less than 2 times. Category types are defined as 1=moderate, 2=strong, 3=severe, 4=extreme. Note that we stored the MHWs characteristics by day instead of by event, as it is usually done, to be used in association with the daily maps of macroevents (see following Section).

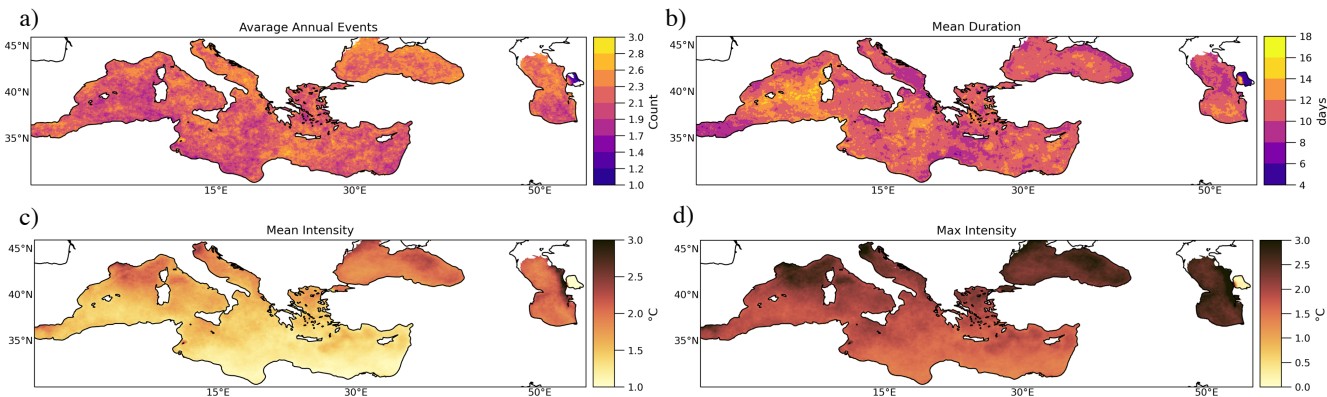

**Figure 5.** MHWs characteristics for each grid cell: a) Average number of annual events, b) Mean Duration, c) Mean Intensity, d) Maximum intensity. The color schemes used in this Figure are produced using the Scientific colour maps package (Crameri et al., 2020).

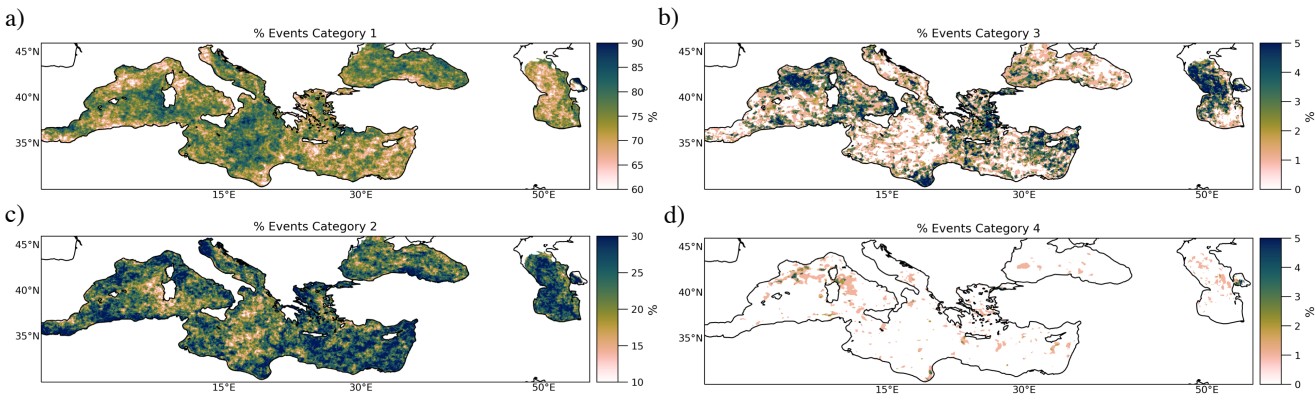

**Figure 6.** Percentage (%) of counted MHWs events in a) Category 1, b) Category 2, c) Category 3, d) Category 4. The color scheme used in this Figure are produced using the Scientific colour maps package (Crameri et al., 2020).

Figure 5 shows the spatial pattern of annual event count, mean duration, mean intensity, maximum intensity of the grid cell

MHWs events over SEWA basins. The Black Sea, the north Caspian Sea, the Adriatic Sea, the Gulf of Lion and the Alboran Sea experienced the most frequent, shortest and most intense MHWs. The majority of the events over SEWA basins belong to the moderate category, reaching the 90% in the Black sea and in the Central Mediterranean Sea (Figure 6). All the basins experienced moderate MHWs, especially the Eastern Mediterranean, the North Adriatic and the Ligurian Seas. 5% of MHWs

are severe over the Eastern part of the Caspian Sea, South of the Gulf of Lion and the Gulf of Sidra. MHWs in category 4 (extreme) are very rare in the study area.

## 3.2 MHWs macroevents detection

The 3D binary maps (i.e. time x longitude × latitude) previously obtained were used to identify spatiotemporally connected marine heatwaves, i.e. grid points that are connected in 3D space in terms of MHWs occurrence (Figure 2) . Specifically, following the approach of Woolway et al. (2021), a connected component analysis is used to identify a connected group of marine heatwave grid cells which, in turn, are considered as part of the same MHW (i.e. a contiguous region simultaneously experiencing a MHW). For this work we used the distributed version of Label function[4] from the python packaged named dask_image.ndmeasure[5]. The label function calculates connectivity of features to their neighbors based on a structuring element matrix establishing the directions in which connectivity is defined. In our case, the inputs of the function are the 3D binary maps chunked by time (the non-zero values in matrices, i.e. MHWs occurrence, in the matrices are counted by the algorithm as features and zero values are considered the background) and the structuring element matrix is orthogonal which means that the features (i.e. grid cells MHWs) are connected in north-south, west-east directions (see help function[6] for details). Since the algorithm works in parallel, first, each chunk is independently labeled (i.e. connection in space). Then, the independent labels are made consecutive and merged along chunks' faces whenever connected (i.e. connection in time). The algorithm returns connected grid cells with a unique label. Each of these unique labels represents a macroevent. After filtering out macroevents with a maximum area lower than 100 km$^2$ (4 grid cells), we found 68068 macroevents over the SEWA region. Similarly, we also applied the filtering to the daily maps of MHWs characteristics.

Figure 7 shows some examples of macroevents detected by our procedure and their evolution in time. Each color identifies a different macroevent. Focusing on the lilac event in Figure 7 we can appreciate the strength of the method. The macroevent starts on 5th of October in three different spots in the Aegean Sea, then it grows spatially reaching its maximum extension on 19th of October exenting over much of the eastern Mediterranean sea. Finally, it decays along the Libyan coast by November, 20th. Meanwhile, other macroevents develop in the basins (e.g. blue label in the Caspian Sea, green label in the Black sea). It is worth clarifying that our MHWs macroevent definition does not consider the physics behind an event. As stated before, the connected component analysis is a statistical method to aggregate grid cells which are experiencing MHWs connected in time and in space. Therefore, macroevents that have been labeled differently not being spatiotemporally connected (e.g. green macroevent and lilac macroevent during 9/11/1983 in Figure 7), could have been triggered by the same causes.

---

[4]https://docs.scipy.org/doc/scipy/reference/generated/scipy.ndimage.label.html#scipy.ndimage.label

[5]http://image.dask.org/en/latest/_modules/dask_image/ndmeasure.html#label

[6]http://image.dask.org/en/latest/_modules/dask_image/ndmeasure.html#label

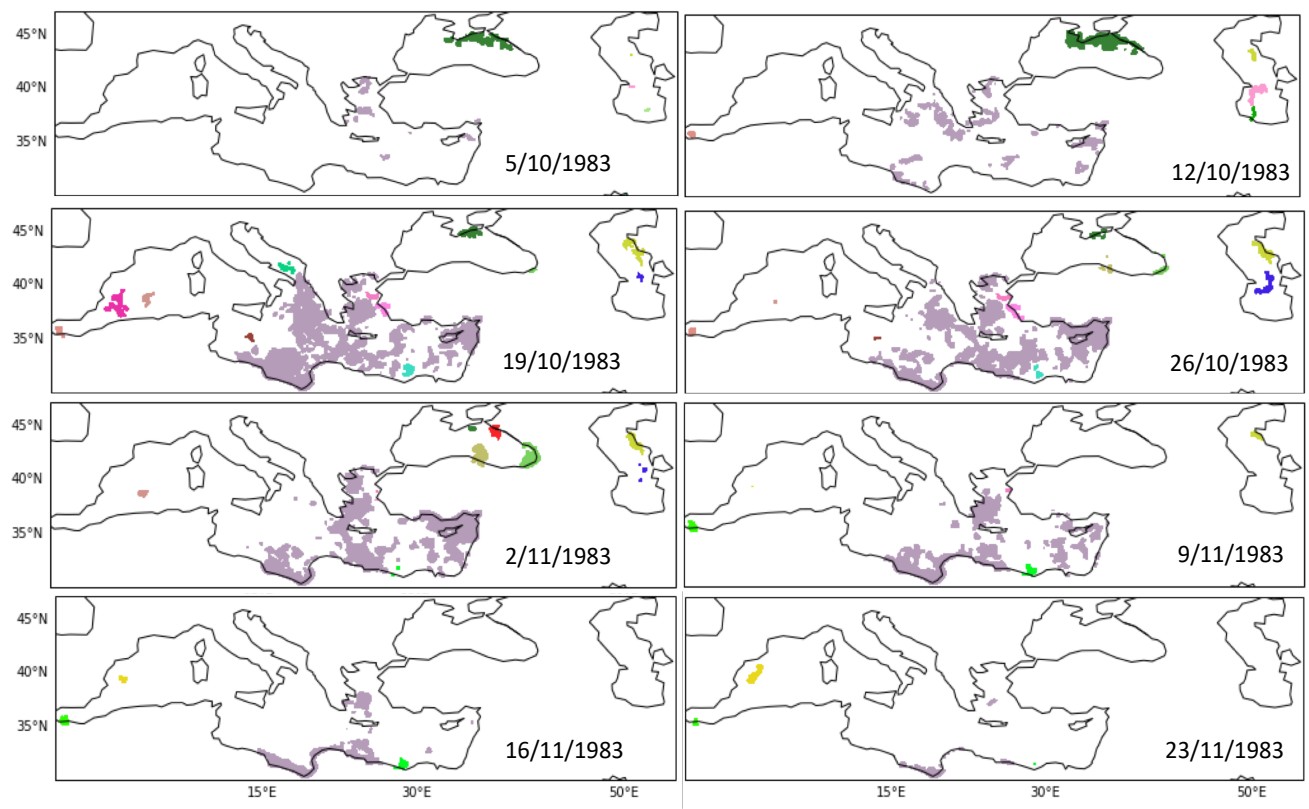

**Figure 7.** MHWs macroevents and their evolution in time, from 5th October 1983 to 23rd November 1983, over the Mediterranean Sea detected by the connected component analysis.

### 3.2.1 Mediterranean MHW of 2003

In order to assess how the proposed method performs in a case of a well known event, we show in details how it detected the 2003 MHW over the Mediterranen Sea (hereinafter MED-MHW-2003). The detected macroevent covered all the western Mediterranean and it lasted 302 days (Figure 8 a and b). Based on the number of active points (i.e. points which simultaneously experienced the labeled MED-MHW-2003, blue line in Figure 8a) and the mean intensity of the active points (orange line) we can distinguish and characterize the evolution of the MED-MHW-2003 in five phases. The spatial patterns of the average mean intensities in Figure 8b are computed, for each grid point, as the sum of the MHWs daily intensities divided by the duration of the phase. The time-series of Figure 8a gives us information about the daily spatial mean of the MED-MHW-2003 intensities, while the spatial patterns shown in Figure 8b teach us about the time mean of intensities during MED-MHW-2003 phases. Table 1 summarizes the characteristics of the MED-MHW-2003 phases. The phases are:

1. Phase 1: lasted from May to June 2003 hitting a maximum daily spatial mean intensity of $2.2°C$. It expanded over the Central Mediterranean sea. The MHW mean intensity over the entire period are around $2°C$ west of Sicily (Figure 8b).

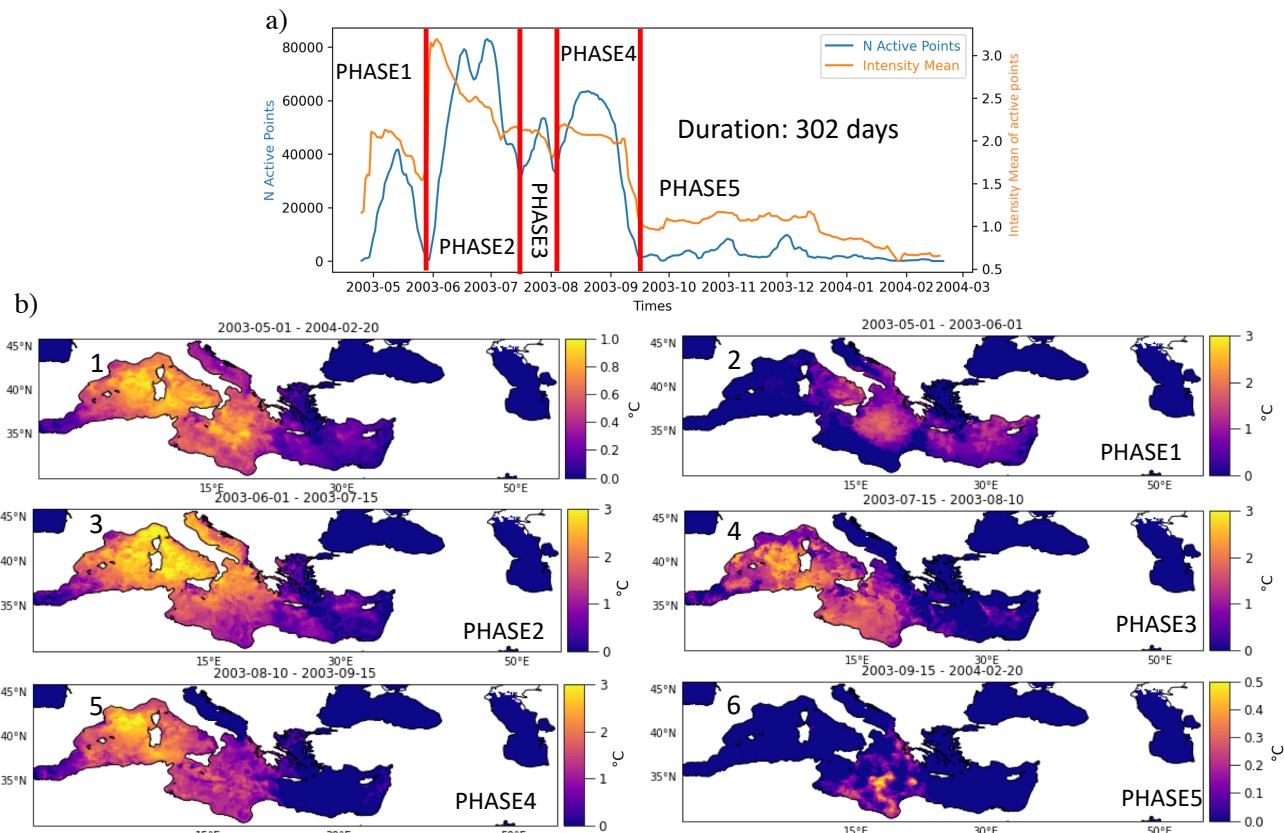

**Figure 8.** a) Active points, i.e. points which simultaneously experienced the labeled MED-MHW-2003 (blue line) and mean intensity of active points (orange) during the 2003 MHW, b) Average of the mean intensities during the MED-MHW-2003 period (1) and during its phases (2,3,4,5,6), computed as the sum of the MHWs daily intensities divided by the duration of the phase.

2. Phase 2: lasted from June to mid-July 2003 hitting a maximum daily spatial mean intensity of 3.2°C (Figure 8a). It expanded over the Western, Central Mediterranean and the Adriatic Seas. The MHW mean intensity over the entire period are around 3-4 °C in the Ligurian Sea, in the Tyrrhenian Sea and on the Adriatic coast (Figure 8b).

3. Phase 3: lasted from mid-July to August 2003 hitting a maximum daily spatial mean intensity of 2.2°C (Figure 8a). It expanded over the Central Mediterranean Sea. The MHW mean intensity over the entire period are around 3°C to the west of Sardinia (Figure 8b).

4. Phase 4: lasted from August to mid-September 2003 hitting a maximum daily spatial mean intensity of 2.2°C (Figure 8a). It expanded over the Western Mediterranean Sea. The MHW mean intensity over the entire period are around 3-4 °C in Gulf of Lion (Figure 8b).

5. Phase 5: lasted from mid-September 2003 to February 2004 hitting a maximum daily spatial mean intensity of 1.2°C (Figure 8a). It expanded over the Central Mediterranean Sea. The MHW mean intensities over the entire period are around 0.5 °C north of the Gulf of Sidra (Figure 8b).

Even though the evaluation of a MHW is strictly dependent on its definition, we compared our results for the MED-MHW-2003 with existing works in the literature. Analyzing SST data from 1985 to 2003 derived from NOAA-AVHRR measurements,Marullo and Guarracino (2003) reported a SST warm anomaly of about 4 °C in June of 2003 in the Gulf of Lion, Ligurian, Tyrrhenian, Northern Ionian and Adriatic Seas similarly to MED-MHW-2003 phase 2 (Figure 8b). Similarly to MED-MHW-2003 phase 3 and 4 shown in Figure 8b, they also observed the persistence of the event in July with weaker
anomalies of about 3 °C, and an intensification in August over the Gulf of Lion and the Ligurian Sea, with recorded anomalies of about 4 °C. In September, as detected in MED-MHW-2003 phase 5, they observed as well a considerable weakening of the event, characterized by anomalies below 1 °C. MED-MHW-2003 phase 1, 2 and 3 are also comparable with the analysis of Grazzini and Viterbo (2003) and Olita et al. (2007). Using the ECMWF forecast products and a regional 3-D ocean model, respectively, they described the event as a rapidly increase of SST anomalies over the Central Mediterranean Basin during
May 2003, which, it intensified (with anomalies of about 3-4 °C) and it expanded, covering the whole Mediterranean with the exception of the Aegean Sea, by the end of July. The data analysed by Sparnocchia et al. (2006) in the Ligurian Sea evidenced the development of a warm anomaly in the SST as reported by MED-MHW-2003 phase 2, 3 and 4. They detected a local SST anomaly up to 2-3 °C, which built up at the end of May and persisted until August 2003.

| Phase | Date | Duration | Area | Max spatial mean intensity | Max time mean intensity |
|-------|------|----------|------|----------------------------|-------------------------|
| 1 | 2003-05-01 2003-06-01 | 31 days | Central Mediterranean Sea | 2.2°C | 2.1°C |
| 2 | 2003-06-01 2003-07-15 | 44 days | Western, Central Mediterranean Seas Adriatic Sea | 3.2°C | 3.6°C |
| 3 | 2003-07-15 2003-08-10 | 26 days | Western, Central Mediterranean Seas | 2.2°C | 3.1°C |
| 4 | 2003-08-10 2004-09-15 | 36 days | Western Mediterranean Sea | 2.2 °C | 3.3°C |
| 5 | 2003-09-15 2004-02-20 | 158 days | Central Mediterranean Sea | 1.2°C | 0.5 °C |

**Table 1.** Summary of the MED-MHW-2003 phases.

### 3.3 SEWA-MHWs dataset summary

Our dataset is composed of daily fields of macroevents and their characteristics (Figure 2). We moved from a grid cell-based dataset to an event-based dataset without losing grid cell information. Moreover, since the drivers and the impacts of MHWs are still not well understood, we included as components of the SEWA-MHWs dataset also some relevant atmospheric parameters taken from ERA5 dataset (Hersbach et al., 2020) to further encourage the use of the SEWA-MHWs dataset. In particular, to promote the study of the drivers and following the work of Sen Gupta et al. (2020), we added the mean sea level pressure, the latent heat, the sensible heat, the incoming solar radiation and the wind speed at 10 m. In addition, air temperature at 2 m is also available to promote studies on the relationship between MHWs and land heatwaves. The area extracted for these meteorological parameters is slightly bigger than the SEWA region, allowing the investigation of remote influences and/or responses of these variables in relationship with MHWs macroevents. All the details, units and name of variables available in SEWA-MHWs dataset are explained in the Zenodo repository (Bonino et al., 2022).

## 4 Scientific Application

### 4.1 Spatial/Agglomerative clustering of MHWs macroevents

In order to highlight the added value of the SEWA-MHWs dataset we studied the largest macroevents out of the 68068 identified by our methodology. In particular, we classified and aggregated the largest MHWs macroevents that share characteristics, taking advantage of statistical clustering methods. Following the work by Stefanon et al. (2012), who identified and classified continental heat waves over Europe during the 1950-2009 period, we devised a clustering procedure to explore similarities and differences among macroevents included in the dataset. Our clustering technique consists of 4 steps:

1. For each identified MHW macroevent, we extracted the maximum area extension reached. We retained MHWs macroevents with area greater than 100 000 km$^2$, which is about 25% of the SEWA basins. We identified 187 largest macroevents out of 68068 macroevents.

2. For each day belonging to one event, we extracted the MHW mean intensity for all the grid points which belong to the macroevent and we set to nan the other grid points. So that, for each macroevent we obtained daily maps of mean intensity.

3. All daily maps belonging to one macroevent are averaged producing 'event maps'. So that, we obtained one event map of mean intensity for each macroevent.

4. An agglomerative hierarchical clustering algorithm (Gordon, 1999) is applied to the event maps. At the initial step, each event map forms a cluster. The two 'nearest' clusters are then merged by pair into a new cluster. We used the cosine distances to measure the distances between clusters. The cosine similarity is defined as the cosine of the angle between vectors (i.e. vectors of event maps), that is, the dot product of the vectors divided by the product of their lengths. The

cosine similarity depends on the angle between vectors, not on their magnitudes. Therefore, the definition of the cosine distance is particularly suited to distinguish between different spatial patterns of the intensities as it tends to increase as the number of grid cells shared by the event maps of two macroevents decreases (see Stefanon et al. (2012) for additional details). We used a Python algorithm to perform the clustering [7].

Different clustering solutions have been considered, ranging from 2 to 14 clusters. Figure 9 shows the average silhouette scores obtained for each of these clustering solutions. The silhouette score is a summary of the distance between a member in a given cluster and the members in the neighboring clusters. It ranges from -1 to 1 and provides a way to assess cluster separation (Kaufman and Rousseeuw, 2009). In particular, a large average silhouette score can be considered as an indication of large separating distances among the resulting clusters, thus implying better clustering results. According to Figure 9a, we concluded

that the optimal number of clusters is 6. In addition, the individual silhouette scores can also be exploited to investigate the quality of this optimal solution (Figure 9b). Different colors correspond to the different clusters, and the thickness of each colored shape identifies the cluster size (i.e. number of samples, in our case event maps, in each cluster). The red dashed line shows the average silhouette score for this solution (in Figure 9a). Most samples (i.e. event maps) have a silhouette score larger than the average score, especially in clusters 4 and 5. This indicates a favorable clustering result, in the sense that most of

the samples seem to be well-separated from the neighboring clusters. Nevertheless, the presence of some values smaller than the average score and negative values suggests that there is some degree of overlap among some of the clusters, with possibly misclassified events.

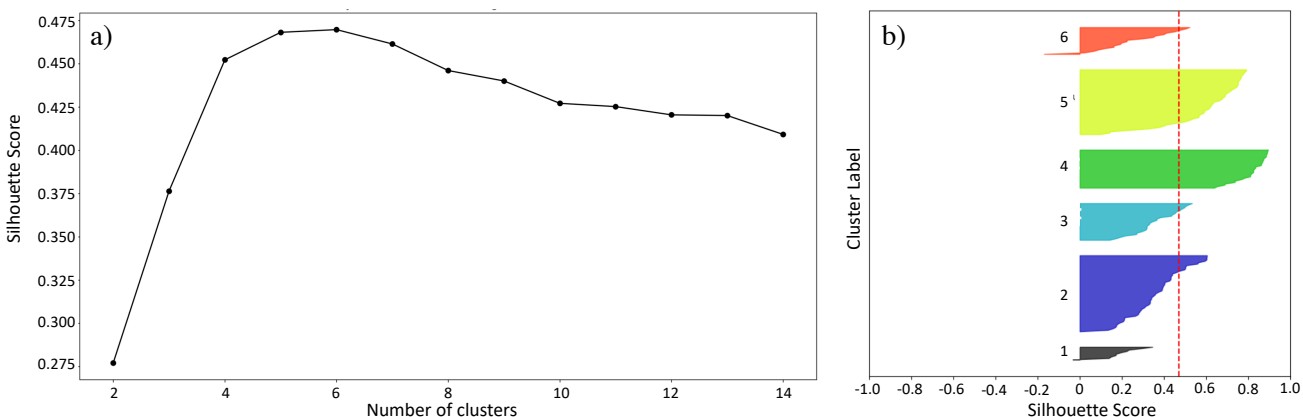

**Figure 9.** a) Average silhouette score for agglomerative clustering performed using cosine distances, b) silhouette analysis for agglomerative clustering performed using cosine distances on data with n clusters = 6.

---

[7]https://scikit-learn.org/stable/modules/generated/sklearn.cluster.AgglomerativeClustering.html

## 4.2  MHWs macroevents clusters and their characteristics

Figure 10 shows the typical marine heatwave patterns for each identified cluster. The patterns of each cluster are represented
by the average of the mean intensity of all the event maps which belong to that cluster. Boxplots for area maxima, duration,
intensity maxima and intensity means for MHW macroevent for each cluster are available in Figure 11, the box shows the
quartiles of the dataset distribution while the whiskers extend to show the rest of the distribution, except for diamonds that are
determined to be "outliers". Moreover, Figure 12 shows the number of MHWs macroevents by season and by decades for each
cluster.

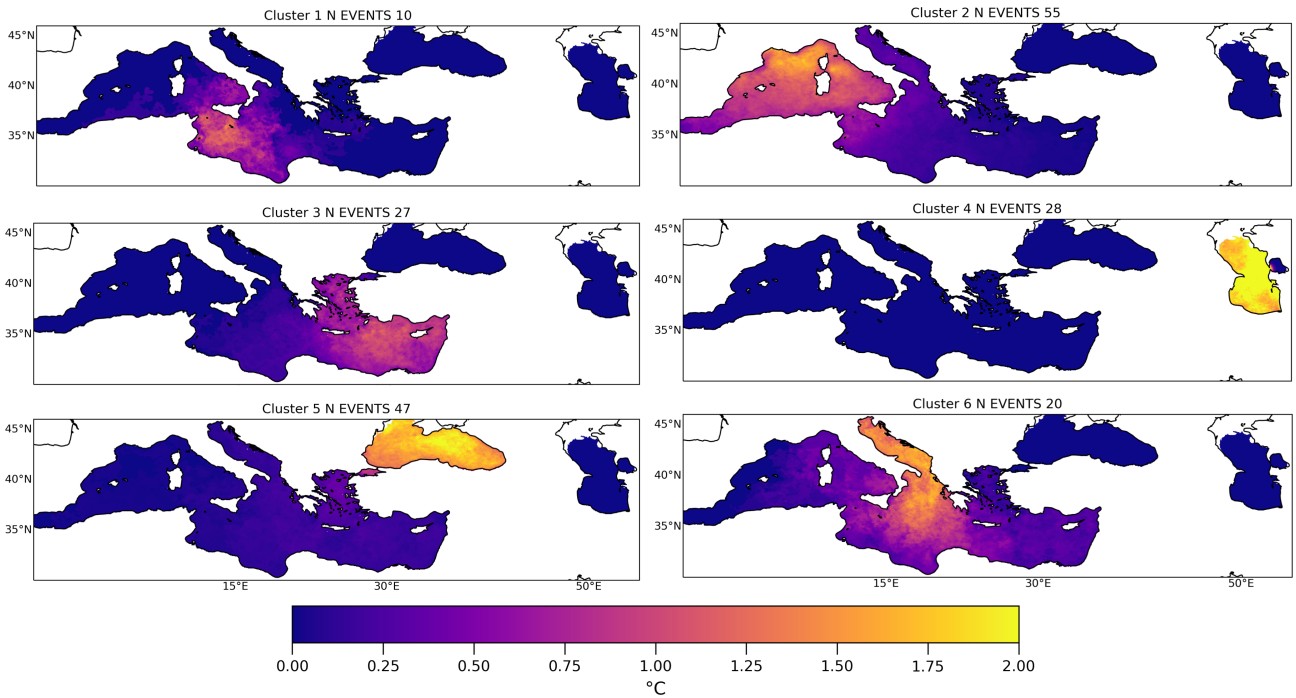

**Figure 10.** Patterns of the clusters represented by the average of the mean intensity of all the event maps of each cluster.

The longest and the largest macroevents, in terms of area maxima, belong to Cluster 2 which spans over the Western-
Mediterranean and the Adriatic Sea. The maximum intensities are located in the Gulf of Lion and the Ligurian Sea and they
decrease magnitude towards the Adriatic Sea. The Aegean Sea, instead, emerges as separate cluster, it counts 27 macroevents
(Cluster 3 in Figure 10). The maximum intensities are located west of Cyprus and they decrease magnitude around the Greek
Archipelago, showing mean intensity of about 1.3°C and maximum intensity of about 2.5°C (Figure 11). The Aegean Sea,
to a lower extent, is also part of other two identified clusters, Cluster 5 and Cluster 6. The former expands in the Black sea,
where it shows its maximum intensity, while the latter includes the Adriatic Sea and the the Ionic sea. Both of these two clus-
ters experienced strong macroevents with maximum intensity of about 3.5°C (Figure 11). Apart from Cluster 6, the Central

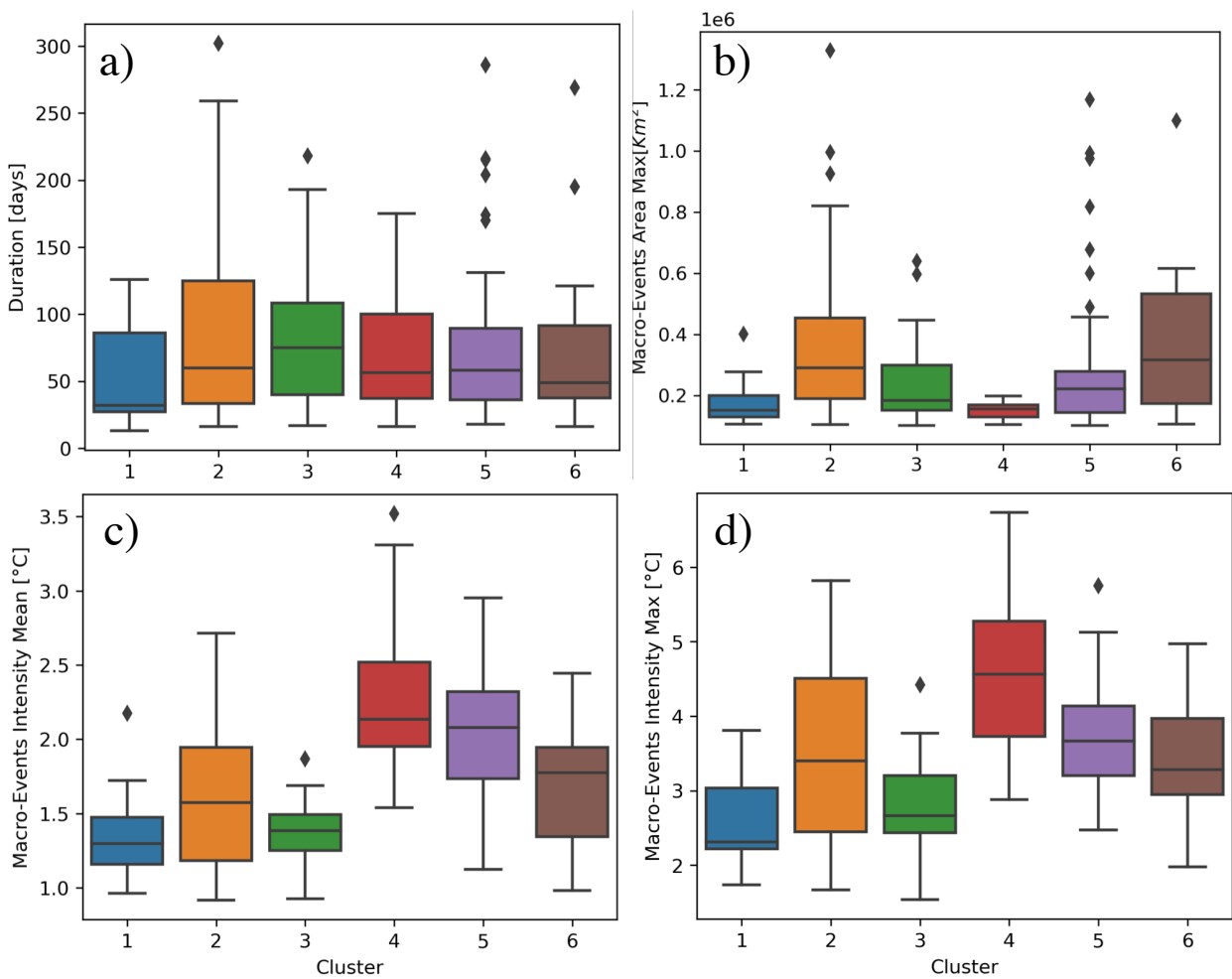

**Figure 11.** Boxplots for a) Duration, b) Area, c) Intensity Mean and d) Intensity Maxima for each cluster.

Mediterranean Sea is also home of the smallest cluster, consisting of 10 macroevents confined from the Strait of Sicily to the eastern Libyan coast (Cluster 1 in Figure 10). The maximum intensities are located south of Sicily, with mean intensity of about 1.3°C and maximum intensity of about 2.5°C (Figure 11). A completely isolated cluster groups the 28 macroevents over the Caspian Sea (Cluster 4 in Figure 10). The maximum intensities are located along the western coast of the basin. The strongest, in term of intensity, macroevents belong to this cluster, reaching 2.3 and 4.5°C of mean and maximum intensity, respectively (Figure 11).

Except for Cluster 6 and Cluster 1, it is interesting to highlight that the majority of the events are during the summer, while the winter produced only few macroevents for each cluster (Figure 12). Moreover, we do not report an increasing number of MHWs during the last six years (2011-2016), as this is reported by Dayan et al. (2022). Actually, the majority of the macroevents in

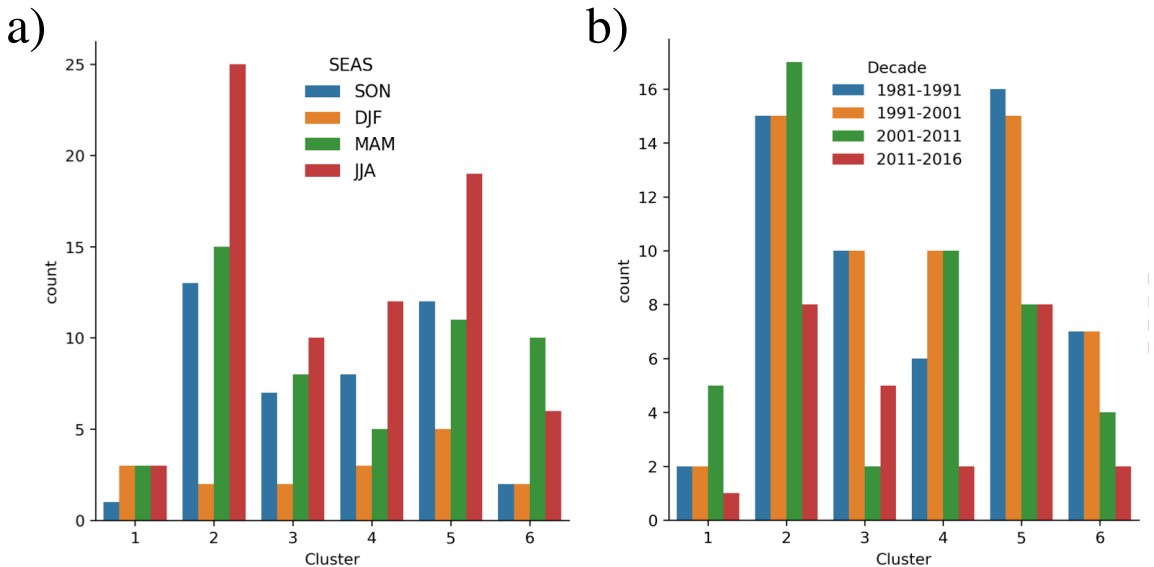

**Figure 12.** a) MHWs macroevents by season for each cluster; b) MHWs macroevents by decades for each cluster.

almost all the clusters occurred during the first two decades of the studied period (Figure 12). This is likely linked to the fact that we considered the trend in the baseline climatology estimation (see section 3.1.1).

Our clustering methodology seems to be effective in distinguishing different spatial patterns of MHWs macroevents over the

SEWA basins. The macroevents result geographically confined to the closed basins (i.e. Caspian Sea and Black Sea) or to the sub-basins of the Mediterranean Sea (e.g. Western Mediterranean Sea), however highlighting some relations between adjacent sea regions (e.g. Adriatic Sea and Aegean Sea, cluster 6).

## 5   Code and Data availability

The SEWA-MHWs dataset, that consists in daily fields of MHWs macroevents, their characteristics, and relevant atmospheric

variables, is stored in the Zenodo archive (Bonino et al., 2022, https://doi.org/10.5281/zenodo.7153255). The MHW detection methodology described in Section 3 is applied to the SEWA region, but it could be, in principle, applied to the global ocean or to other basins. Morevorer, the SEWA-MHWs dataset is inevitably linked to the ESA CCI SST dataset; indeed, all the datasets that are produced from or reuse high-quality data depend on the data used to generate them. Even though the routines are computational and time demanding, we provide scripts to rerun the method over other regions or using other and updated SST

datasets. We provide the code to detect MHWs and their characteristics (MHWs_stl.ipynb), the code to generate the MHWs macroevents (SEWA_LABEL.ipynb), and the code to filter out the smallest macroevents (MHWs_filter.ipynb). The codes are

also available in the Zenodo repository. Please refer to the dataset description in Bonino et al., 2022 for any details on the netcdf files and on the codes.

## 6 Summary and Outlook

In this work we presented a dataset of Marine Heatwaves macroevents and their characteristics over the Southern Europe and Western Asia basins during the 1981-2016 period, named SEWA-MHWs dataset. We obtained the dataset by analyzing observed SST provided by the European Space Agency, the ESA CCI SST v2.1 dataset (Merchant et al., 2019). Briefly, we defined MHWs in each 5x5 km grid point of the ESA CCI SST dataset, then, using a connected component analysis, we aggregated the spatiotemporally connected MHWs (i.e. grid points that are connected in 3D space in terms of marine heatwave

occurrence) in order to obtain MHWs macroevents. As a result, the SEWA-MHWs dataset, consists in daily field of MHWs macroevents and their characteristics.

The methodological framework used to build SEWA-MHWs dataset is the novelty of this study with respect to the existing literature (e.g. Darmaraki et al., 2019; Sen Gupta et al., 2020; Oliver et al., 2021). Firstly, the detected MHWs in the ESA CCI SST dataset are relative to a time-varying baseline climatology, which considers both the trend and seasonal variability to

355 mimic the changing of the climate mean state. Secondly, the connected component analysis allow us to aggregate the MHWs connected in time and in space and to pass from a the grid cell-based dataset to an event-based dataset without loosing high resolution (i.e. grid cell) information. This approach, differently from the previous studies, provides the time evolution of the event at the basin scale. Even though the evaluation of a MHW is strictly dependent on its definition, we demonstrated that our method is effective in detecting MHWs macroevents. The well known MHW of 2003 in the Mediterranean Sea is comparable

with the records (e.g. Olita et al., 2007; Sparnocchia et al., 2006; Marullo and Guarracino, 2003; Grazzini and Viterbo, 2003). To the best of our knowledge, the SEWA-MHWs dataset is the first effort in the literature in archiving extreme hot sea surface temperature macroevents. The advantages of the availability of a MHWs macroevents dataset are to avoid waste of computational and/or time resources to process SST data to detect MHWs, and to build a consistent framework which would increase comparability among MHWs studies. As Pastor and Khodayar (2022) and Sun et al. (2022) suggested in very recent papers,

the scientific community should focus on establishing a universal definition of MHWs events which does not rely only on the grid cell definition. Besides the consistency ensured among MHWs studies that will use SEWA-MHWs dataset, SEWA-MHWs dataset also provides a ready-to-use dataset to be compared to other studies which apply different MHW definitions. On top of that, Pastor and Khodayar (2022) also suggest that it should be mandatory to introduce some spatial limitations in the study of MHWs, especially when targeting impacts. Our users could, as we did for the statistical clustering, filter out macroevents based

on their needs. The SEWA-MHWs dataset can be used for many scientific applications. For instance, we efficiently clustered the biggest SEWA-MHWs macroevents that share common features characteristics in order to report and to characterize typical spatial patterns of MHWs over SEWA basins. Indeed, the employed clustering method was able to distinguish different spatial patterns of MHWs macroevent.

The SEWA-MHWs dataset is also suitable for regional and coastal MHWs studies, due to its high resolution, and, it is expandable to all the ocean basins to provide a global coverage. Moreover, the synergistic use of SEWA-MHWs dataset with other model outputs and observation data could help to fill the knowledge gaps about the drivers and the marine ecosystems impacts of these extreme events. Recently, compound events have become of particular interest, i.e., when conditions are extreme for multiple potential ocean ecosystem stressors such as temperature and chlorophyll (Gruber et al., 2021, Le Grix et al., 2021). On top of that, this synergistic use of SEWA-MHWs dataset with other datasets could facilitate the building of prediction scheme using, for example, deep machine learning approaches. These techniques need large and high resolution datasets to be trained and tested.

In a broader perspective, the novel science resulting from the aforementioned exploitation of the SEWA-MHWs dataset could be transferred into solutions and advanced decision support systems for society.

*Author contributions.* GB and SM conceived the study. GB, SM, GG and MM discussed and defined the methodological framework. GB and MM set up the code for STL decomposition and for the MHWs macroevents detection. GB performed the clustering, all the analysis and wrote the manuscript. GB, SM, GG, MM interpreted the results. SM and GG revised and contributed to improving the manuscript.

*Competing interests.* The authors declare that they have no conflict of interest.

*Acknowledgements.* This research has been funded by the European Space Agency (ESA) as part of the FEVERSEA Climate Change Initiative (CCI) fellowship (ESA ESRIN/Contract No. 4000133282/20/I/NB). We acknowledge the CMCC Foundation for providing computational resources.

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
