# Peer review of "Southern Europe and Western Asia Marine Heat Waves (SEWA-MHWs): a dataset based on macro events"

_Earth System Science Data, 2022_

## Author Response (AR1)

**Dear Referee #1,**

we would like to thank you for the careful reading of the manuscript and the constructive comments that substantially helped to improve and clarify the paper. Answers to all your comments are detailed hereafter. Corrections to the English grammar were adopted in the revised version of the manuscript according to the reviewer's recommendations, but are not reported or discussed here. All authors agree with the modifications made to the manuscript. The comments by the referee are reported in bold followed by our response (in blue). The text added to the revised manuscript is reported in italic font. The line numbers reported in the answers referred to the revised manuscript. The revised manuscript that includes track changes is also provided in pdf format.

In the following answers, we use 'Figure' to identify the figures in the updated manuscript and we use 'Plot' to identify the figures in this document.

**General comments:**

The manuscript describes a dataset of MHW events built on ESA CCI SST. The dataset is geographically limited to Southern Europe and Western Asia, even if the SST dataset is global. The approach is rooted in Hobday et al 2016 framework, but introducing refined statistical methods for the detection of marine heatwaves.

I have enjoyed reading the manuscript; it is well written, the length is just about right, it is clear and concise. The description of the dataset is appropriate. I easily downloaded the dataset and was able to inspect it without problems.

From a methodological point of view, the manuscript is certainly interesting for the community. The proposed methodology to take into account shifts toward warmer climate is something we have to deal with and the way the authors tackle this is worth reading. The clustering analysis of macro events is valuable, although the authors did not attempt to associate driving synoptic (atmospheric) conditions to each clusters, something that was the reason for the clustering in Stefanon et al 2012. However, this is likely beyond the mere description of the dataset required by ESSD.

**Major comments:**

Even if I find the methodological part very interesting, I am a little bit doubtful about the relevance of the dataset itself. The reduced geographical boundaries limit the number of potential users, while it would have been straightforward to run the methodology on the global dataset. The STL-method needs to be re-run every time ESA CCI SST is updated (e.g., including recent years), otherwise the dataset gets quickly outdated. The fact that this dataset is inevitably linked to a specific SST dataset may limit its relevance. Scientists may want to rerun the methodology but on different or newer SST datasets. The significance of the dataset itself, in my opinion, is thus limited.

We thank the referee for the comment on the methodology. Pastor and Khodayar (2022) in a very recent paper highlighted the need to establish a universal definition of MHWs events, which does not rely only on the grid-cell definition, to ensure intercomparison of results.

Moreover, in another very recent paper, Sun et al. 2022 suggest that defining an MHW event under the spatiotemporal framework provides a more fruitful description of MHW characteristics. Our dataset is an attempt to aggregate single events by defining the macroevents, and building a consistent framework that would increase comparability among MHWs studies.

Since in the interactive discussion with the editors we have already addressed some of the points raised by the referee, in the following paragraphs we just report a modified and updated version of that discussion (https://doi.org/10.5194/essd-2022-343-AC1).

Based on the review criteria of ESSD journal (https://www.earth-system-sciencedata.net/peer\_review/review\_criteria.html), we think that the dataset has its significance. The significance criterion is divided into three sub-criteria: uniqueness, usefulness, completeness.

Since SEWA-MHWs is the first effort in literature in archiving extreme hot sea surface temperature macroevents, we think that the SEWA-MHWs dataset could be considered as unique. The advantages of the availability of a MHWs macroevents dataset are to avoid waste of computational and/or time resources to process SST data to detect MHWs.

The dataset can be considered useful as well, because, alone or in combination with other data sets, can be used in future interpretations, for the comparison to model output or to verify other numerical experiments or observations. In particular, as stated before, it could help to fill the knowledge gaps about the drivers and the marine ecosystems impacts of these extreme events. For example, the dataset could be used in the analysis of compound events, i.e., when conditions are extreme for multiple potential ocean ecosystem stressors such as temperature and chlorophyll (Gruber et al., 2021, Le Grix et al., 2021). Moreover, our attempt to provide a complete dataset in a consistent framework would increase comparison among MHWs studies that will use SEWA-MHWs dataset and, on top of that, SEWA-MHWs dataset provides a ready-to-use dataset to be compared to other studies which apply different MHW definition, without waste of computational and/or time resources. Lastly, the SEWA-MHWs dataset can be considered complete as it covers one semi-closed basin and two closed basins for the entire period of ESA CCI SST v2.1.

Nevertheless, as the referee pointed out, we are aware that the dataset has some limitations:

1) The reduced geographical boundaries limit the number of potential users. This is certainly true. However, besides the fact that the method is too computationally demanding to be applied globally, not all the datasets arise as global. Moreover, for climate change and in particular, for this specific phenomenon, the Mediterranean is a hot spot (Garrabou et al., 2009, Giorgi et al. 2006, Cramer et al., 2018, Pastor et al., 2020, Garrabou et al., 2022, Pastor and Khodayar, 2022). Mediterranean SST satellite data analysis has shown a positive trend in the last 40 years with progressively higher temperatures extending longer within the annual cycle but also with more frequent and intense SST extremes (Pastor et al., 2020). This SST warming has been linked to the increasing occurrence of MHWs in the last decades in the Mediterranean (Ciappa, 2022). In addition, the Mediterranean Sea is generally recognized as an exemplar model to assess the ecological effects of climate change (Garrabou et al 2022, Cramer et al., 2018). Although the Mediterranean Sea represents only 0.32% of the total

volume of the oceans, its unique geomorphological history led to a markedly high level of biodiversity with 7%–10% of all known marine species and a large proportion of endemic species (Bianchi & Morri, 2000; Coll et al., 2010). Marine heatwaves caused unprecedented biological impacts in the Mediterranean Sea (Garrabou et al 2022, Cramer et al., 2018; Marbà et al., 2015; Rivetti et al., 2014,), seriously affecting marine biodiversity (Juza et al., 2022). Therefore, the MHWs Mediterranean community is increasing constantly and new projects that could benefit from our work have already started (e.g. CareHeat, https://eo4society.esa.int/projects/careheat/). The relevance of MHW events occuring in the Mediterranean Sea is also shown by news recently reported on various international newspapers or websites, e.g. https://www.lemonde.fr/en/environment/article/2022/07/30/marine-heat-waves-meandeadly-fate-for-large-number-of-mediterranean-flora-and-fauna 5991965 114.html, https://www.esa.int/Applications/Observing\_the\_Earth/Mediterranean\_Sea\_hit\_by\_marine heatwave, https://www.mercator-ocean.eu/actualites/marine-heatwaves-mediterraneansummer-2022/, https://www.reuters.com/business/cop/mediterranean-marine-heatwavesthreaten-coastal-livelihoods-2022-11-13/, https://news.mongabay.com/2022/09/mindblowing-marine-heat-waves-put-mediterranean-ecosystems-at-grave-risk/.

Finally, it is worth stressing that the application of the methodology described in the paper could be extended beyond the Mediterranean, and, in principle, could be applied to the global ocean. Therefore, we provide scripts to rerun the method over other regions. In particular, we provided the code to detect MHWs and their characteristics (MHW\_stl.ipynb), the code to generate the MHWs macroevents (MED\_LABEL.ipynb), and the code to filter out the smallest macroevents (MED\_filter.ipynb). The scripts are available in the Zenodo repository (Bonino et al., 2022).

2) The dataset misses recent years. This is certainly true, but to build our dataset we used the most updated version of the ESA CCI SST. Nevertheless, we provide scripts to rerun the method on the updated ESA CCI SST v3 dataset. In particular, we provided in the Zenodo repository (Bonino et al., 2022) the code to detect MHWs and their characteristics (MHW\_stl.ipynb), the code to generate the MHWs macroevents (MED\_LABEL.ipynb), the code to filter out the smallest macroevents (MED\_filter.ipynb).

3) The dataset is inevitably linked to a specific SST dataset. This is certainly true, but we think that this is an intrinsic characteristic of all the datasets that are produced from or reuse highquality data. Inevitably, they depend on the data used to generate them. However, it is worth mentioning that the methodology proposed in the paper is independent of the specific SST dataset. Thus, it can be easily rerun on new SST datasets. To facilitate this, we provided the code to detect MHWs and their characteristics (MHW\_stl.ipynb), the code to generate the MHWs macroevents (MED\_LABEL.ipynb), and the code to filter out the smallest macroevents (MED\_filter.ipynb) in the Zenodo repository (Bonino et al., 2022).

To clarify point 1 we added the following text in the introduction at lines 67-73:" We have focused on SEWA basins because they represent a well known "Hot Spot" region for climate change (Giorgi, 2006) and, in particular, for this specific phenomenon (Garrabou et al., 2009, Giorgi et al. 2006, Cramer et al., 2018, Pastor et al., 2020, Garrabou et al., 2022, Pastor and Khodayar, 2022, Ciappa et al., 2022). Marine heatwaves caused unprecedented biological impacts, especially in the Mediterranean Sea (Garrabou et al 2022, Cramer et al., 2018; Marba

et al., 2015; Rivetti et al., 2014,), seriously affecting marine biodiversity (Juza et al., 2022). The Mediterranean Sea is recognized as an exemplary model for assessing the ecological and biological impacts of climate change (Garrabou et al 2022, Cramer et al., 2018). "

For points 2 and point 3, we added the following paragraphs to the data and code availability section, which we moved before the summary and outlooks: "The SEWA-MHWs dataset, that consists in daily fields of MHWs macroevents, their characteristics, and relevant atmospheric variables, is stored in the Zenodo archive (Bonino et al., 2022). The MHW detection methodology described in Section 3 is applied to the SEWA region, but it could be, in principle, applied to the global ocean or to other basins. Morevorer, the SEWA-MHWs dataset is inevitably linked to the ESA CCI SST dataset; indeed, all the datasets that are produced from or reuse high-quality data depend on the data used to generate them. Even though the routines are computational and time demanding, we provide scripts to rerun the method over other regions or using other and updated SST datasets. We provide the code to detect MHWs and their characteristics (MHWs\_stl.ipynb), the code to generate the MHWs macroevents (SEWA\_LABEL.ipynb), and the code to filter out the smallest macroevents (MHWs\_filter.ipynb). The codes are also available in the Zenodo repository. Please refer to the dataset description in Bonino et al., 2022 for any details on the netcdf files and on the codes"

Moreover, following the suggestion of the reviewer#2 Peter Minnett, we added some relevant atmospheric variables from ERA5 dataset to make the dataset more attractive to potential users. In particular to encourage the study of the drivers, following the work of Sen Gupta et al. 2019, we added the mean sea level pressure, the latent heat, the sensible heat, the incoming solar radiation and the wind speed at 10 m. Moreover, following Peter Minnett's suggestion, we also add the air temperature at 2m to encourage studies on heatwaves relationship.

This is clarified at lines 258-266:" Moreover, since the drivers and the impacts of MHWs are still not well understood, we included as components of the SEWA-MHWs dataset also some relevant atmospheric parameters taken from ERA5 dataset (Hersbach et al., 2020) to further encourage the use of the SEWA-MHWs dataset. In particular, to promote the study of the drivers and following the work of Sen Gupta et al. (2020), we added the mean sea level pressure, the latent heat, the sensible heat, the incoming solar radiation and the wind speed at 10 m. In addition, air temperature at 2 m is also available to promote studies on the relationship between MHWs and land heatwaves. The area extracted for these meteorological parameters is slightly bigger than the SEWA region, allowing the investigation of remote influences and/or responses of these variables in relationship with MHWs macroevents. All the details, units and names of variables available in SEWA-MHWs dataset are explained in the Zenodo repository(Bonino et al., 2022)."

**Specific remarks:**

1) Geographical names cited in the text should be shown somewhere, to help readers unfamiliar with the region

We agree with the reviewer, and we added the geographical names in Plot 1 (Figure 1a in the manuscript).

Plot 1: Mean SST climatology detected by STL with geographical names. Blue circle identifies the Western Mediterranean location of the time-series shown in Figure 4.

2) Section 3.2 the definition of macro event is >64km2 (4 pixels), while the definition of macro event in 1 is >100000 km2. I think this is confusing and the first definition is not really about "macro events", maybe the authors should call the filtering out of few pixels in other way.

We thank the reviewer for the comment. Actually, the definition of macroevent is >64km2 (4 pixels). We then filtered out macroevents with an area less than 100000 km2 to study the largest macroevents and to cluster them. We clarified this point at lines 269-271:" In order to highlight the added value of the SEWA-MHWs dataset we studied the largest macroevents out of the 68068 identified by our methodology. In particular, we classified and aggregated the largest MHWs macroevents that share characteristics, taking advantage of statistical clustering methods..... We identified 187 largest macroevents out of 68068 macroevents."

3) Figure 3: it is not clear to me if the % is on counted events in a category or total days within a category

We thank the reviewer, we modified the caption of the new Figure 6: "Percentage (%) of counted MHWs events in a) Category 1, b) Category 2, c) Category 3, d) Category 4"

4) The title of Cleveland et al 1990 in the references is incomplete. Also, title of figure 2D should be "Remainder", not Reminders.

We thank the reviewer. We corrected the reference and the title.

5) Figure 4C and 4D: 0 in the colorbar should be in white color

We agree with the review and we revised Figure 6 (Plot 2).

Plot 2: Percentage (%) of counted MHWs events in a) Category 1, b) Category 2, c) Category 3, d) Category 4

**6) I have the visual feeling that fig6a and fig 6b are not consistent. For example, the average intensity in phase 5 in fig 6a is always >0.5 while in fig 6b it is hardly above 0.5. Am I wrong?**

We agree with the reviewer. Actually, Figure 6a and 6b show two different intensities. The time-series in Figure 6a shows the daily mean intensity of the active points i.e. points which simultaneously experienced the labeled MED-MHW-2003 in that day. Whereas Figure 6b shows the MHW mean intensity for each grid point, computed as the sum of the MHWs daily intensities, which could be zeros during some days, divided by the duration of the phase. We clarified this point in the caption of Figure 8 (Figure 6 in the old manuscript): " a) Active points, i.e. points which simultaneously experienced the labeled MED-MHW-2003 (blue line) and mean intensity of active points (orange) during the 2003 MHW, b) Average of the mean intensities during the MED-MHW-2003 period (1) and during its phases (2,3,4,5,6), computed as the sum of the MHWs daily intensities divided by the duration of the phase." We also add this clarification at lines 221-224: "The spatial patterns of the average mean intensities in Figure 8b are computed, for each grid point, as the sum of the MHWs daily intensities divided by the duration of the phase. The time-series of Figure 8a gives us information about the daily spatial mean of the MED-MHW-2003 intensities, while the spatial patterns shown in Figure 8b teach us about the time mean of intensities during MED-MHW-2003 phases. Table 1 summarizes the characteristics of the MED-MHW-2003 phases." Moreover, we expanded and revised the description of the phases and we added Table 1, as suggested by the reviewer#2 Peter Minnett, to summarize the characteristics of the MED-MHW-2003 phases.

| Phase | Date                     | Duration | Area                                                | Max spatial mean intensity | Max time mean intensity |
|-------|--------------------------|----------|-----------------------------------------------------|----------------------------|-------------------------|
| 1     | 2003-05-01
2003-06-01 | 31 days  | Central Mediterranean Sea                           | 2.2°C                      | 2.1°C                   |
| 2     | 2003-06-01
2003-07-15 | 44 days  | Western, Central Mediterranean Seas
Adriatic Sea | 3.2°C                      | 3.6°C                   |
| 3     | 2003-07-15
2003-08-10 | 26 days  | Western, Central Mediterranean Seas                 | 2.2°C                      | 3.1°C                   |
| 4     | 2003-08-10
2004-09-15 | 36 days  | Western Mediterranean Sea                           | 2.2 °C                     | 3.3°C                   |
| 5     | 2003-09-15
2004-02-20 | 158 days | Central Mediterranean Sea                           | 1.2°C                      | 0.5 °C                  |

Table 1. Summary of the MED-MHW-2003 phases.

**7) Line 267 pg 13: "during the last decades (2011-2016)". 5-years period cannot be decades ... also, "as instead reported in Dayan et al., 2022": you may want to specify why you get different results**

We agree with the reviewer, we revised the sentence at lines 325-328: "Moreover, we do not report an increasing number of MHWs during the last six years (2011-2016), ... Actually, the majority of the macroevents in almost all the clusters occurred during the first two decades of the studied period (Figure 12). This is likely linked to the fact that we considered the trend in the baseline climatology estimation (see section 3.1.1).".

**REFERENCES**:**

Bianchi, C. N., & Morri, C. (2000). Marine biodiversity of the Mediterranean Sea: situation, problems and prospects for future

---

## Referee Report (RR1)

**Review of the revised version of the paper: "Southern Europe and Western Asia Marine Heat Waves (SEWA-MHWs): a dataset based on macro events", by Bonino et al.**

Reviewer 1 found the paper clear, concise, and well written, and the dataset produced easy to download and inspect. He/her also appreciated the methodological part, based on the clustering of macro events, but had a major concern about the significance of the dataset, considering it limited by the geographic restrictions, and by the fact that the dataset is linked to a specific SST and misses recent years. I think that the authors have clearly and convincingly dissipated this concern, both in the answer to the referee and introducing appropriate changes in the text, to further highlight the significance of the new dataset. In my opinion, even if the dataset was restricted to the Mediterranean region only, it would nonetheless be significant for a wide, multidisciplinary scientific community, which is involved in the study of this climatic hotspot and in the protection of the basins' rich and delicate ecosystems.

The authors have also taken into consideration the other points raised by Reviewer 1 and have carefully addressed all the thoughtful and helpful comments by Reviewer 2, Dr. Peter Minnett, introducing changes that have improved the paper, and the dataset itself.

I therefore recommend that the revised manuscript be accepted for publication in ESSD.